# Impact of deep brain stimulation of the subthalamic nucleus on natural language in patients with Parkinson's disease

**Felicitas Ehlen**[1,2]*, **Bassam Al-Fatly**[1], **Andrea A. Kühn**[1,3,4,5,6], **Fabian Klostermann**[1,3]

**1** Department of Neurology, Humboldt-Universität zu Berlin and Berlin Institute of Health, Charité-Universitätsmedizin Berlin, Berlin, Germany, **2** Department of Psychiatry and Psychotherapy, Jüdisches Krankenhaus Berlin, Berlin, Germany, **3** Berlin School of Mind and Brain, Humboldt-Universität zu Berlin, Berlin, Germany, **4** Neurocure Cluster of Excellence, Humboldt-Universität zu Berlin and Berlin Institute of Health, Charité-Universitätsmedizin Berlin, Berlin, Germany, **5** Bernstein Center for Computational Neuroscience, Humboldt-Universität zu Berlin and Berlin Institute of Health, Charité—Universitätsmedizin Berlin, Berlin, Germany, **6** Deutsches Zentrum für Neurodegenerative Erkrankungen, Berlin, Germany

* felicitas.ehlen@charite.de

**Data Availability Statement:** Our data are stored in the public repository G-Node GIN (doi: 10.12751/g-node.mb83iq).

## Abstract

### Background

In addition to the typical motor symptoms, a majority of patients suffering from Parkinson's disease experience language impairments. Deep Brain Stimulation of the subthalamic nucleus robustly reduces motor dysfunction, but its impact on language skills remains ambiguous.

### Method

To elucidate the impact of subthalamic deep brain stimulation on natural language production, we systematically analyzed language samples from fourteen individuals (three female / eleven male, average age 66.43 ± 7.53 years) with Parkinson's disease in the active (ON) versus inactive (OFF) stimulation condition. Significant ON-OFF differences were considered as stimulation effects. To localize their neuroanatomical origin within the subthalamic nucleus, they were correlated with the volume of tissue activated by therapeutic stimulation.

### Results

Word and clause production speed increased significantly under active stimulation. These enhancements correlated with the volume of tissue activated within the associative part of the subthalamic nucleus, but not with that within the dorsolateral motor part, which again correlated with motor improvement. Language error rates were lower in the ON vs. OFF condition, but did not correlate with electrode localization. No significant changes in further semantic or syntactic language features were detected in the current study.

### Conclusion

The findings point towards a facilitation of executive language functions occurring rather independently from motor improvement. Given the presumed origin of this stimulation effect

**Funding:** This study was supported by the German Research Foundation (KI-1276/5 in Clinical Research Group 247) and the Open Access Publication Fund of Charité – Universitätsmedizin Berlin. Author BA has received a Doctoral Research Grant from the German Academic Exchange Service - DAAD. AAK has received honoraria for advisory activities from Medtronic, Boston Scientific, Abbott, Ipsen and Teva. FK has received honoraria for advisory activities from CSL Behring and Theranexus. The funders had no role in study design, data collection and analysis, decision to publish, or preparation of the manuscript.

**Competing interests:** The authors have read the journal's policy and have the following competing interests: AAK has received honoraria for advisory activities from Medtronic, Boston Scientific, Abbott, Ipsen and Teva. FK has received honoraria for advisory activities from CSL Behring and Theranexus. This does not alter our adherence to PLOS ONE policies on sharing data and materials. There are no patents, products in development or marketed products associated with this research to declare.

within the associative part of the subthalamic nucleus, this could be due to co-stimulation of the prefrontal-subthalamic circuit.

## Introduction

Apart from disabling motor symptoms, Parkinson's disease (PD) can lead to wide-ranging non-motor symptoms [1]. These involve cognitive processes [2] which typically also affect functions relevant to language such as memory, set shifting, flexibility, planning, and the integration of semantic networks [3–5]. The majority of PD patients thus develop language symptoms including impaired fluency [6–9] (for reviews see [10]) with an increase in speech hesitations [11], and slower speech initiation [12]. On the syntax level, symptoms may comprise global impairments in sentence generation [13, 14] (cf. [12]), decreased syntactic complexity [11] (cf. [12]), and a specific decline in verbs [15–17]. Moreover, an abnormally low informational content [14, 18] and impaired pragmatic language production have been described [18, 19]. Finally, comprehension deficits regarding complex sentences (e.g., [12, 20–25]) and motor speech dysfunction [12, 26] can add to disease-related communication problems.

Deep Brain Stimulation (DBS) of the subthalamic nucleus (STN) is a well-established therapeutic option for patients suffering from complex PD motor symptoms despite optimized drug treatment [27] (for a review see [28]). One proposed mechanism of action is a modulation of excessive beta frequency synchronization within the cortico-basal circuits [29–31]. The STN is embedded in the inhibitory "indirect" loop [32], while also being monosynaptically connected with frontal motor and prefrontal cognitive areas [33, 34] via the excitatory "hyperdirect" pathway [35]. In this central position, the nucleus is expected to enable a protraction of premature answers [34, 36] and a modulation [37] of inputs from cortical projection regions. Further, cross-modal signal integration is a postulated STN function, given marked overlap at the border zones [34, 38, 39] of its dorsolateral motor, ventromedial associative, and rostral limbic sections [32]. Cognitive STN functions, including language processing, have accordingly been suggested to mainly involve inhibitory response control [5, 17, 33, 40–42], response selection [4, 43], and action sequencing [44]. However, although STN DBS applied to the dorsolateral section exerts strong prokinetic motor effects, its impacts on cognition are ambiguous with the majority of studies suggesting a DBS-related acceleration of cognitive speed [36, 40, 41, 45–47], at the expense of premature responses leading to reduced accuracy [40, 41, 45, 48, 49].

Specifically regarding effects of STN DBS on natural language production, only a few studies have been conducted and delivered equivocal results: [50–54] whereas reduced hesitations, paraphasias, and errors together with improved lexical retrieval were interpreted as a functional recovery [55], other studies showed vastly unaltered language functions [53, 56] or compromised grammatical capacities [52].

Since this heterogeneity could relate to differences between language tasks as well as the impact of variable stimulation fields [57], the current study aims to analyze a comprehensive range of linguistic parameters in spontaneous language samples from persons with PD with respect to the volume of tissue activated (VAT) by STN DBS. Under the premise of mainly parallel effects on motor and language functions, we hypothesized that STN DBS should primarily counteract excessive inhibition, resulting in enhanced fluency, sentence generation, and informative content. At the same time, an increase in language errors could be expected to result from DBS-induced disinhibition.

## Participants and methods

### Participants

Fourteen individuals (three female, average age 66.43 ± 7.53 years, average disease duration 13.79 ± 4.63 years, average DBS treatment duration of 3.14 ± 1.95 years) treated with bilateral, constant voltage-driven STN DBS participated in the study. All participants met the UK Brain Bank Criteria for PD and were right-handed native German-speakers. The individual STN DBS settings had been optimized with respect to clinical parameters during a one week stay on our specialized ward for DBS patients following the implantation and during regular (i.e. quarterly) consultations in our specialized outpatients' department. Compatible with other clinical cohorts (e.g., [58]), most of our participants were treated with monopolar stimulation as it requires lower stimulation intensity. Bipolar stimulation had been chosen if an individual developed adverse effects arising from the comparably large volume of tissue activated by monopolar stimulation [59]. An overview of individual demographic, disease and DBS-related data is provided in Table 1.

Exclusion criteria were a previous or current history of brain disease other than PD, including all psychiatric disorders such as depression, psychosis, or apathy as well as dementia (assessed by the Parkinson Neuropsychometric Dementia Assessment; PANDA [60]) or unintelligible speech irrespective of the stimulation condition.

To obtain data for ON-OFF comparisons, all participants were examined in two separate sessions, i.e. in the STN DBS ON and STN DBS OFF condition at an interval of two months and systematically alternating order. Examinations in the ON condition were carried out under therapeutic stimulation parameters that had been stable for at least two months prior to the assessment. For the OFF condition, stimulation was switched off at least thirty minutes before the examination. The individually optimized antiparkinsonian medication remained stable and the timing of the assessments was planned individually to ensure the best clinical "medication-ON" state (starting time at about 11 a.m. for most participants; the last medication intake was protocolled.

All participants were recruited from the outpatient clinic for movement disorders of the Charité University Hospital Berlin. They gave written informed consent to the study protocol approved by the ethics committee of the Charité–Universitätsmedizin Berlin (protocol number EA2/047/10). The research was conducted in accordance with current guidelines and the Declaration of Helsinki.

### STN DBS implantation

Implantation of tetrapolar, cylindrical DBS electrodes (Medtronics, model 3387; contact height: 1.5 mm, diameter: 1.27 mm) into the STN had been performed by stereotactic surgery based on preoperative MRIs using atlas coordinates, intraoperative microelectrode recordings, and macroelectrode stimulation. Correct localizations had been confirmed by post-operative MRIs. All operations had been carried out by the same neurosurgical team at the Charité University Hospital.

### Electrode localization and calculation of volume of tissue activated

Electrode localization and calculation of VAT was performed in all participants except case 8 due to missing postoperative imaging. DBS leads were localized with Lead-DBS open access MATLAB software (www.lead-dbs.org, version 2 [61]). To estimate of the VAT, Lead-DBS incorporates pre-existing models (first proposed by [62–65]) as well as tractography algorithms in order to transition to the global volume of modulation [61]. The localization process

**Table 1. Overview of demographic, disease and DBS-related data.**

**A**

| Pt. | Age (ys) | Sex | Body Side of Onset | School (ys) | Hoehn &Yahr | Disease Dur. (ys) | DBS Dur. (ys) |
|---|---|---|---|---|---|---|---|
| 1 | 55 | M | Right | 10 | 2.5 | 10 | 2 |
| 2 | 62 | M | Right | 10 | 2 | 15 | 6 |
| 3 | 61 | F | Left | 10 | 3 | 18 | 2 |
| 4 | 67 | M | right | 10 | 3 | 16 | 5 |
| 5 | 77 | M | right | 12 | 4 | 19 | 2 |
| 6 | 75 | M | right | 11 | 3 | 15 | 4 |
| 7 | 72 | M | left | 12 | 2 | 15 | 3 |
| 8 | 55 | F | right | 10 | 2 | 15 | 7 |
| 9 | 62 | F | right | 8 | 3 | 22 | 8 |
| 10 | 70 | M | right | 10 | 2 | 11 | 4 |
| 11 | 74 | M | right | 8 | 2.5 | 14 | 2 |
| 12 | 73 | M | right | 8 | 2 | 4 | 0.5 |
| 13 | 58 | M | left | 13 | 2 | 10 | 4 |
| 14 | 69 | M | left | 10 | 2 | 9 | 0.5 |

**B**

| Pt. | Voltage (V) | | Impulse Width (μs) | | Frequency (Hz) | | Impedance (Ω) | | Polarity | |
|---|---|---|---|---|---|---|---|---|---|---|
| | left | right | Left | right | left | right | Left | right | Left | right |
| 1 | 3.4 | 2.9 | 90 | 60 | 90 | 90 | 818 | 1145 | Bi | Mo |
| 2 | 2.6 | 3.8 | 90 | 60 | 90 | 90 | 1035 | 1010 | Bi | Mo |
| 3 | 2.2 | 3 | 60 | 90 | 90 | 90 | 604 | 594 | Mo | Mo |
| 4 | 3.9 | 2.4 | 90 | 90 | 160 | 160 | 812 | 962 | Mo | Mo |
| 5 | 3.7 | 3.8 | 60 | 60 | 130 | 130 | 500 | 956 | Mo | Mo |
| 6 | 3.8 | 3.8 | 60 | 60 | 90 | 90 | 500 | 459 | Mo | Mo |
| 7 | 2.9 | 2.9 | 60 | 60 | 130 | 130 | 766 | 766 | Mo | Mo |
| 8 | 3.5 | 4.4 | 60 | 60 | 80 | 80 | 459 | 283 | Mo | Mo |
| 9 | 1.7 | 1.3 | 60 | 60 | 130 | 130 | 577 | 441 | Mo | Mo |
| 10 | 2.7 | 2.7 | 60 | 60 | 130 | 130 | 712 | 712 | Mo | Mo |
| 11 | 3.5 | 2.1 | 60 | 60 | 130 | 130 | 848 | 523 | Bi | Mo |
| 12 | 2.2 | 1 | 60 | 60 | 130 | 130 | 1083 | 2365 | Mo | Mo |
| 13 | 2.1 | 2 | 60 | 60 | 130 | 130 | 550 | 523 | Mo | Mo |
| 14 | 0.5 | 0.5 | 60 | 60 | 130 | 130 | 2457 | 2084 | Mo | Mo |

The table provides an overview of the participants included in the language analysis (n = 14) regarding demographic, disease, and DBS related data; bi: bipolar; dur.: duration; f: female; H&Y: Hoehn & Yahr Stage (DBS ON); m: male; mo: monopolar; Pt.: participant code

has been described in detail in earlier studies [65, 66] and shall briefly be outlined: postoperative images (CT or MRI) were co-registered to the preoperative MRI using either Advanced Normalization Tools (ANTs; http://stnava.github.io/ANTs/ [67]) for postoperative CT scans or the Statistical Parametric Mapping (SPM12) toolbox (http://www.fil.ion.ucl.ac.uk/spm/software/spm12/) for postoperative MRI scans. Brain-shift correction was implemented in a further step to adjust for any bias caused by possible operation-related pneumocephalus. ANTs normalization pipeline was applied to warp preoperative and postoperative acquisitions to MNI (ICBM 2009b NLIN asymmetric) space. DBS electrodes were automatically reconstructed in MNI space, followed by manual refinement if needed (electrode localization in the xyz-space is provided in the S1 Table). Next, the VAT was modelled around each clinically

used active contact. Therefore, each participant's clinical stimulation parameters were used to calculate a volume conductor model of the DBS electrode and surrounding tissue (in analogy to the procedure described in [65]). For this purpose, a tetrahedral volume mesh based on the surface meshes of DBS electrodes and subcortical nuclei was generated using the Iso2Mesh toolbox included in Lead-DBS. Regions that were neither filled with conductive/insulating electrode material nor with gray matter were assigned to white matter. Gray matter nuclei were defined by the DISTAL atlas [68]. Conductivities of 0.14 S/m and 0.33 S/m were assigned to the white and grey matter, respectively (cf. [69]). We used values of 108 S/m for the platinum/iridium contacts and 10–16 S/m for the insulated parts of the electrodes. A simulation of the potential distribution resulting from the DBS followed based on the volume conductor model. As boundary condition we used the voltage applied to the active electrode contacts. For monopolar DBS, the surface of the volume network represented the anode. By derivation of a finite element method solution (toolbox simbio/FieldTrip incorporated in lead dbs software), the gradient of the potential distribution was determined, resulting in a piecewise continuous gradient. The extent and shape of the activated tissue volume were defined by setting the gradient as a threshold value above the frequently used value of 0.2 V/mm (cf. [38, 65, 70, 71]). Next, each participant's bilateral VATs were overlapped with the motor and associative area of the STN as provided by the DISTAL atlas [68] (see Fig 1). For each STN segment, the size of the overlap cluster was normalized by taking its ratio to total VAT size. To assess whether the results are dependent on the 0.2 V/mm threshold, we performed a control analysis that did not include a heuristic/arbitrary threshold, but instead used the full electric field (see S1 Fig).

## Motor and neurocognitive assessment

We used the motor section (i.e., part III) of the Unified Parkinson's Disease Rating Scale (*UPDRS*; minimum score = 0; maximum score = 108; higher scores indicate stronger symptom severity) to assess motor symptom severity and the *PANDA* [60] (minimum score = 0; maximum score = 30; higher scores indicate stronger symptom severity) to evaluate cognitive functions such as working memory, executive functions, and verbal fluency. As a part of the UPDRS motor score, speech intelligibility was scored on the given scale (i.e., 0, 1, 2, 3, or 4 points indicate no, slight, mild, moderate, or severe speech problems, respectively [73]).

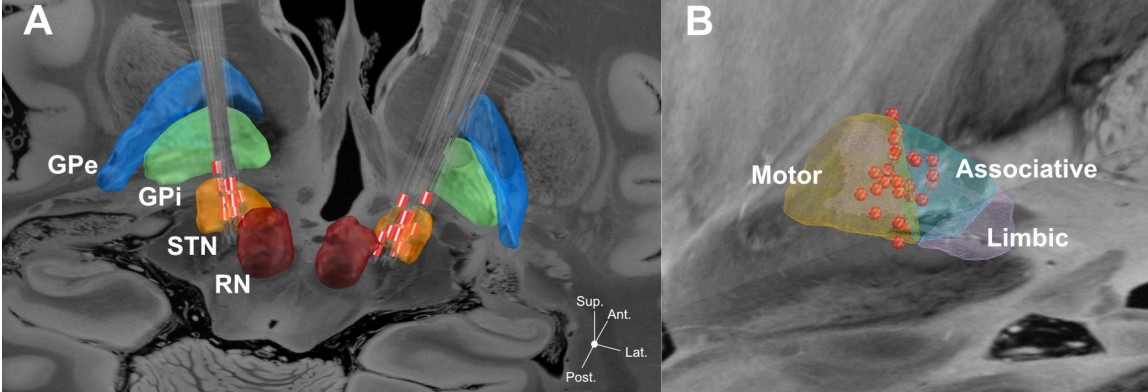

**Fig 1. Localization of DBS electrodes within the STN. A:** DBS electrodes localizations from thirteen participants in relation to the STN. Electrodes have been reconstructed in MNI space. Red contacts represent clinically active contacts. **B:** Spatial distribution of active contacts in relation to different STN segments (from DISTAL atlas). All 26 active contacts (red spheres) are projected cumulatively onto the left hemisphere (right contacts have been flipped). Subcortical structures including STN segments were defined from DISTAL subcortical atlas implemented in Lead-DBS [68]. Backdrop is from the high resolution 100 micron postmortem MRI in Edlow et al. [72]. Please note that due to missing post-operative MRI the data from participant no. 8 did not enter the localization analysis. GPe: globus pallidus externus, GPi: globus pallidus internus, RN: red nucleus, STN: subthalamic nucleus.

## Linguistic analysis

**Spontaneous language samples.**   Participants were comfortably seated in a sound-proof chamber. Semi-structured interviews were carried out by an interviewer trained in psychological interviewing. To obtain spontaneous language samples of at least 60 seconds, one out of six predefined open questions was posed in a randomized order per participant and session (i.e., related to i. school days, ii. working life, iii. parents, iv. origin, v. vacation, or vi. hobbies; cf. [54, 56, 74]). If the answer did not yield a sufficiently long monologue, the interviewer formulated further questions by either relating to the current answer or by choosing a different question out of the six above. All interviews were digitally recorded (software: Audacity 1.3.13-beta, microphone: the t.bone MB 88U Dual). For further analysis, a monologue of about sixty seconds was excerpted from each language sample. The end of the interviewer's question was defined as starting point of each monologue.

**Transcription.**   Interview transcriptions were conducted according to the guidelines developed for the "Aachener Sprachanalyse" [75] using the computer software 'Praat' (version 6.0.29).

**Linguistic analysis, parameters.**   With the aim to control for global comparability between monologues recorded in both conditions, we first assessed *monologue duration* (in seconds; excluding interruptions by the interviewer) and the *articulation rate* (syllables / s, excluding pauses > 200 ms). Next, to explore typical language difficulties described for PD patients, we analysed i. the *word production rate* (words / s), ii. the total duration of linguistic pauses (i.e., pauses corresponding to the grammatical structure), iii. the total duration of non-linguistic pauses (i.e., pauses interrupting the natural flow of speech), iv. the *language error rate* (ratio language errors / total number of words), v. the *clause production rate* (clauses / s), and vi. *sentence complexity* (ratio complex / simple sentences; i.e., sentences including either at least one subordinate clause or more than one main clause vs. sentences including only one main clause). To estimate flexibility of word use, we additionally analysed vii. the *rate of open class words* (i.e. words conveying semantic information [76] including full verbs, nouns, adjectives, and modal adverbs which altogether can be extended by acquiring new words [77]) and viii. the *type-token ratio* (i.e., the proportion of distinct lexemes).

The above parameters have generally been established in similar study designs [51, 53, 55, 74, 75, 78–82]. To additionally evaluate semantic and syntactic measures not represented by the above parameters, we performed an analysis of stylistic devices. A detailed list of all assessed stylistic devices is provided in the S2 Table. In summary, the production rate of *semantic stylistic devices* and *syntactic stylistic devices* was each expressed as the sum value of the subcategories of semantic figures (i.e., '*addition*', '*omission*', '*transposition*', '*permutation*', '*other*') and syntactic figures (i.e., '*addition*', '*omission*', '*transposition*', '*other*'), respectively, per second.

Detailed data is publicly available at https://doi.org/10.12751/g-node.mb83iq.

**Linguistic analysis, procedure.**   To obtain the above described parameters, the monologues were subjected to manual linguistic analysis including the assessment of: number of words; number and duration of linguistic and non-linguistic pauses; number and type of language errors (i.e., grammatical, lexical, phonetic, contextual, stylistic, idiomatic, pragmatic, logic); number and type of word classes (as defined by the German standard dictionary Duden [83]); number and type of constituents (28 subcategories); morphosyntactic categorizations of verbs, nouns, pronouns, adjectives/adverbs including word complexity; number and type of clauses (i.e., 6 subcategories of main clauses; 18 subcategories of subordinate clauses; sentence equivalents); number of sentences, and number and type of semantic and syntactic stylistic devices.

## Statistical analysis

The *UPDRS* motor score, *PANDA* score, and the interval between last medication intake and assessment onset were compared in the ON vs. OFF condition using two-tailed paired sample *t*-tests. The UPDRS speech item was compared using the Wilcoxon signed-rank test.

Regarding the linguistic analyses, we first compared *monologue duration* and *articulation rate* in the ON vs. OFF condition using two-tailed paired sample *t*-tests. Next, the above described linguistic parameters i. *word production rate*, ii. the total duration of linguistic pauses, iii. the total duration of non-linguistic pauses, iv. *language error rate*, v. *clause production rate*, vi. *sentence complexity*, vii. *rate of open class words*, and viii. *type-token ratio* were compared in the ON vs. OFF condition using two-tailed paired sample *t*-tests (except for *sentence complexity* where the Wilcoxon signed-rank tests was applicable). To determine the false discovery rate of these eight ON-OFF comparisons, we applied the Benjamini–Hochberg procedure setting α to 0.05 (i.e., critical value $\leq ((i/m)^* 0.05)$; with $i$ = *p*-value rank; $m$ = number of comparisons; [84]).

*Production rates of semantic* and *syntactic stylistic devices* were assessed using two separate repeated measures ANOVAs (each with the within-subjects factor 'stimulation' with two levels). The first contained all categories of semantic figures (i.e., 'addition', 'omission', 'transposition', 'permutation', 'other'; each expressed as sum value of their subcategories per second), the second contained all categories of syntactic figures (i.e., 'addition', 'omission', 'transposition', 'other'; each expressed as sum value of their subcategories per second).

To identify relationships between stimulation fields and changes in linguistic parameters, we calculated ON-OFF difference values for each significantly changed linguistic parameter and performed linear Spearman correlations with the ratio of the VAT overlap with the motor and the associative area of the STN. Since electrodes were switched on and off bilaterally rather than separately, overlap ratios were averaged for left and right hemispheric VATs.

The same significant ON-OFF difference values were furthermore related to ON-OFF difference values of the *UPDRS* motor score and the *PANDA* score using linear regression analyses.

Effect sizes of ON-OFF differences were estimated using Cohen's *d* ($> 0.2$: small; $> 0.5$: medium; $> 0.8$: large effect size [85]) for *t*-tests and partial $\eta^2$ ($> 0.10$: small; $> 0.25$: medium; $> 0.40$: large effect size [86]) for pairwise comparisons resulting from the ANOVAs for stylistic devices. All statistical analyses were performed in IBM SPSS Statistics (version 25).

## Results

The *UPDRS* motor score was significantly lower in the STN DBS ON vs. OFF condition (ON: 19.571 ± 7.871 points; OFF: 38.000 ± 13.156 points; $p < 0.001$; Cohen's *d*: 1.700). The *PANDA* score did not change significantly (ON: 23.643 ± 3.225 points; OFF: 22.500 ± 3.653 points; $p = 0.345$; Cohen's *d*: -0.332). Speech was generally well intelligible in both stimulation conditions (ON median: 1 point; OFF median: 1 point; $p = 0.656$; $z < 0.001$). The interval between the last medication intake and the beginning of the language assessment did not differ significantly between the OFF vs. ON condition (ON: 1.64 ± 1.434 hours; OFF: 2.286 ± 1.369 hours; $p = 0.089$; Cohen's *d*: 0.459).

With respect to general comparability of the monologues, no significant ON-OFF differences were found regarding *monologue duration* (ON: 65.841 ± 9.438 s; OFF: 65.376 ± 16.913 s; $p = 0.922$; Cohen's *d*: -0.034) or *articulation rate* (ON: 5.102 ± 0.954 syllables / s; OFF: 4.730 ± 0.673 syllables / s; $p = 0.251$; Cohen's *d*: -0.450).

As presented in Table 2, statistical analysis of ON-OFF differences indicated a significantly higher *word production rate* and *clause production rate* in the ON vs. OFF condition and a

**Table 2. Linguistic parameters in the STN DBS ON and OFF condition.**

| | DBS ON | | DBS OFF | | | | |
|---|---|---|---|---|---|---|---|
| | Mean | SD | Mean | SD | *p*-value | Cohen´s *d* | BH |
| **Word production rate** (Words / s) | 2.049 | 0.43 | 1.727 | 0.469 | 0.004* | -0.718 | 0.007 |
| **Clause production rate** (Clauses / s) | 0.306 | 0.09 | 0.223 | 0.091 | 0.007* | -0.926 | 0.014 |
| **Language error rate** (Language errors / total number of words) | 0.095 | 0.05 | 0.142 | 0.057 | 0.018* | 0.897 | 0.021 |
| **Rate of open class words** (Open class words / total number of words) | 0.404 | 0.05 | 0.377 | 0.076 | 0.148 | -0.434 | 0.029 |
| **Type-token ratio** (Distinct lexemes / all lexemes) | 0.621 | 0.05 | 0.611 | 0.068 | 0.546 | -0.171 | 0.043 |
| **Sentence complexity** (Complex / simple sentences) | 0.744 | 1039 | 0.583 | 0.55 | 0.861 | -0.193 | 0.05 |
| **Total duration of non-linguistic pauses** (s) | 6.518 | 3.149 | 10.020 | 6.221 | 0.049 | 0.710 | 0.025 |
| **Total duration of linguistic pauses** (s) | 16.076 | 7.942 | 16.273 | 9.330 | 0.946 | 0.023 | 0.050 |

Paired *t*-tests were applied to investigate ON-OFF differences (except for 'sentence complexity' where the Wilcoxon signed-rank test was applied). ON-OFF differences were considered as significant if the *p*-value was below the Benjamini-Hochberg critical value (BH); according values are marked with an asterisk.

lower *language error rate* in the DBS ON condition. No significant differences were indicated regarding the *rate of open class words*, the *type-token ratio*, *sentence complexity*, and the *total duration of linguistic* or *non-linguistic pauses*.

Regarding stylistic devices, ANOVAs showed a generally higher *production rate of semantic stylistic devices* in the ON condition, which failed to reach the level of significance (ON: .249 ± 0.078 devices / s; OFF: .158 ± 0.077 devices / s; *p* = 0.061; Cohen's *d*: -1.180) and despite very large standard deviations indicated a significant increase in figures of permutation (ON: 0.053 ± 0.045 figures / s; OFF: 0.017 ± 0.019 figures / s; *p* = 0.016 (after Bonferroni correction); partial $\eta^2$ = 0.489). An increase in *syntactic stylistic devices* did not reach the level of significance (ON: 0.196 ± 0.096 devices / s; OFF: 0.144 ± 0.082; devices / s; *p* = 0.063; Cohen's *d*: -0.585) with none of the single syntactic figures changing significantly.

Electrode localizations within the STN as illustrated in Fig 1 (collapsed for the left and right STN) indicated 20 out of a total of 26 electrodes to be located within the dorsolateral motor area and six within the ventromedial associative area.

The observed increase in the *clause production rate* in the ON vs. OFF condition was strongly and significantly positively correlated with VAT overlap with the associative area of the STN (*r* = 0.779, *p* < 0.001; see Fig 2A top). This correlation was on a slightly weaker, yet significant level also present for the *word production rate* (*r* = 0.498, *p* = 0.047; see Fig 2A bottom). Furthermore, improvement in the *UPDRS* motor score in the ON condition was positively correlated with the average ratio of VAT overlap with the motor area of the STN (*r* = 0.592, *p* = 0.026; see Fig 2B). On the contrary, no significant correlation was indicated between the increase in the *clause* (*r* = 0.051, *p* = 0.420) or *word* (*r* = -0.152, *p* = 0.312) *production rates* and the ratio of overlap with the motor area or between the improvement in the *UPDRS* motor score and the overlap with the associative area of the STN (*r* = -0.306, *p* = 0.158).

We identified no significant correlation between the observed reduction in the *language error rate* in the ON vs. OFF condition with VAT overlap with the associative (*r* = -0.08, *p* = 0.406) or the motor area of the STN (*r* = 0.400, *p* = 0.085).

We found no significant relationship between ON-OFF difference values of the *UPDRS* motor score and *clause* ($r^2$ = 0.121; *p* = 0.224) or *word* ($r^2$ = 0.103; *p* = 0.264) *production rates* or language error rates ($r^2$ = 0.005; *p* = 0.815) and no significant relationship between difference values of the *PANDA* score and *clause* ($r^2$ = 0.030; *p* = 0.553) or *word* ($r^2$ = 0.005; *p* = 0.806) *production rates* or language error rates ($r^2$ = 0.055; *p* = 0.419).

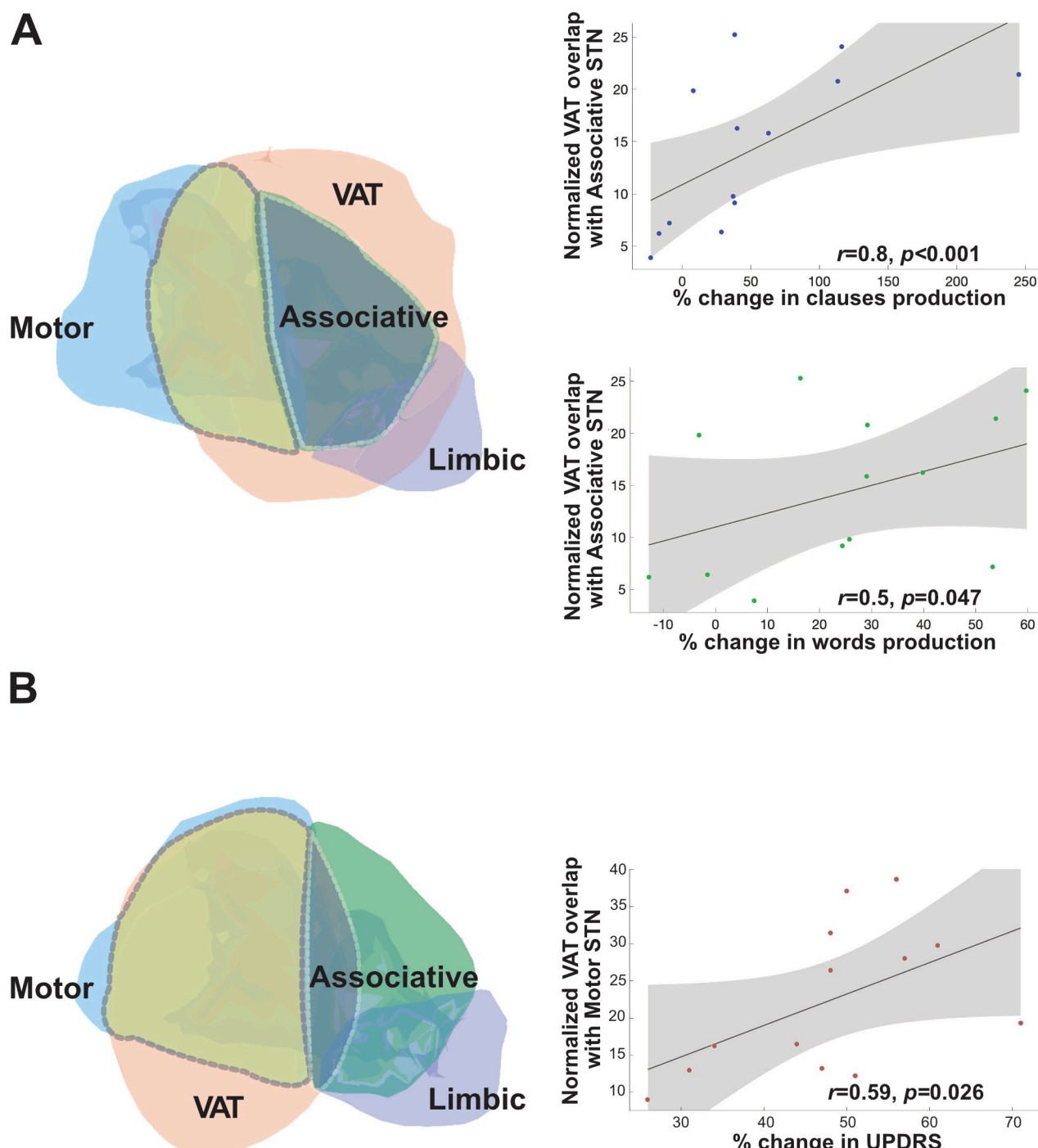

**Fig 2. Correlations between VAT overlap with associative vs. motor STN and language vs. motor changes. A:** VAT overlap (light orange area) with the associative area (green area) of the STN is depicted on the left; scatter diagrams on the right indicate correlations between VAT overlap with the associative STN and the increase in clause (right, bottom) and word (right, center) production. **B:** VAT overlap (light orange area) with the motor area (blue area) of the STN depicted on the left and the scatter diagram on the right indicate a correlation between VAT overlap with the motor STN and UPDRS improvement.

## Discussion

The present study showed a significant increase in clause and word production rates as well as a decrease in the language error rate with subthalamic DBS in PD patients. Interestingly, we found a strong positive correlation between the increase in clause production and VAT overlap with the associative area of the STN. The same but weaker correlation was found for word production. Conversely, no significant relationship was found between the two parameters and the ratio of VAT overlap with the STN motor area. We identified no significant correlation between language error rates and the VAT overlap with the associative or motor area of the STN. Despite large effect sizes, changes in the use of stylistic devices and the total duration of non-linguistic pauses did not reach the level of significance. No significant ON-OFF differences were detectable for sentence complexity, the use of open class words or diverse lexemes, or the total duration of linguistic pauses, each showing rather low effect sizes. In the following, these results shall be discussed in detail.

The main finding of increased clause and word production rates seems compatible with the idea that the STN accelerates cognitive speed [40, 41, 45–47] and particularly modulates language on the level of response selection [4, 43] and inhibitory control [5, 17]. The current results thus appear to reflect a procedural acceleration of functions typically impaired in PD patients, i.e., sentence production [13, 14] and fluency [6–9]. Importantly, this enhancement seemed vastly unrelated to the DBS-induced motor improvement, which correlated with the VAT overlap with the dorsolateral STN.

Thus, DBS appears to unfold differential effects on motor and cognitive functions if electrodes are placed in the hypothesized overlap zone between the motor and associative portion of the STN, as was mostly the case in the present cohort (see Fig 1). This resembles a dissociation between STN DBS effects on motor vs. non-motor functions described for cognitive switching [87], executive functions [36], and impulsiveness [88], which was attributed to a relatively ventral electrode position ([87] but cf. [88]) or a specific recruitment of fibers of the hyperdirect pathway [36, 88]. The present results furthermore seem consistent with an earlier finding of enhanced verbal fluency performance induced by relatively antero-medial STN DBS in the largely overlapping cohort [89]. They, however, extend this finding to the clinically more meaningful production of natural language. By argumentum e contrario, it is further of note that reduced verbal output was reported for tissue stimulation dorsally [90] or laterally [91] to the actual target area and that stimulation directly within the associative STN portion led to lexical and syntactic decline in a previous small-sample study [78]. This underscores the importance of electrode placement within the anterior part of the dorsolateral STN to obtain motor improvement by stimulation of the motor section, while also exerting acceleratory effects on language, probably by co-activation of the associative section closely connected to prefrontal cortical areas. In this framework it is of interest that experimental low frequency stimulation of the ventral STN at 5 Hz improved response control in a Stroop task [92] and dorsolateral stimulation at 4 Hz appeared to normalize the response latency in an interval timing task [33], which deteriorated at 10 Hz stimulation [93]. The authors therefore proposed a functional coupling between the prefrontal cortex and the STN in the delta/theta frequency range, which could be involved in cognitive control processes, including cue processing, response inhibition, working memory recruitment, and attention [33, 92]. Having said that, improved cognitive parameters also indicated a possible activation of fronto-subthalamic connections when stimulated at 10 Hz [94]. Overall, these results are compatible with the idea of a relevant role of the prefrontal subthalamic signaling pathway, which can be disturbed or compensated by co-stimulation of the associative part of the STN or its connective fibers, depending on stimulation parameters, localization and the cognitive requirements of the task type.

Given the typical increase of commission errors under STN DBS [40, 41, 45, 48, 49, 88, 95–97], the current finding of reduced language errors may seem surprising. First, it should be noted that our cumulative parameter comprised diverse linguistic errors that were distributed heterogeneously among the participants (see S3 Table). Nevertheless, the result suggests differential effects of STN DBS on accuracy in the context of naturalistic language production compared to specific cognitive tasks: while stimulation appears to have positive effects on language accuracy if cognitive load is not manipulated, increased error rates in challenging tasks (e.g., [36, 41, 96, 97]) were interpreted as interference of STN DBS with inhibitory control functions, e.g., if increased speed, decision making, or the need for no-go strategies imposed a significant cognitive stress on the individual. Recent studies have demonstrated a relationship between corresponding errors and the connectivity strength between subthalamic VAT and prefrontal areas (including the supplementary motor area, the ventromedial prefrontal cortex, and the inferior frontal gyrus; [36, 88]) Furthermore—consistent with the hypothesis of the physiological "stopping" function of STN—seemingly beneficial effects of preoperatively strong frontostriatal network connectivity on low impulsiveness and disinhibition could be reversed by STN DBS [88]. Future studies could further explore DBS effects on language accuracy by employing tasks to assess the effects of increased cognitive demands on different types of language errors.

Ultimately, we could not identify a correlation between the reduction of language errors and the VAT overlap with the cognitive or motor part of the STN. This could be due to their heterogeneity and possibly to a complex involvement of co-stimulated structures, beyond those investigated here. It seems therefore important to consider further sources of differential STN DBS effects. Against the background of a relevant density increase of interneurons from anterodorsolateral to posteroventromedial, Lévesque and Parent (2005) proposed their specific involvement in the processing of associative information such as motivation, anticipation, planning and integration within the ventromedial STN [98]. Additionally, two different types of glutamatergic principal neurons prevail in the dorsolateral vs. ventromedial STN, suggesting differential functions for sensorimotor vs. associative information processing [98]. Thus, the increase in word and clause production observed here seems to occur along the gradual transition of cell composition from dorsolateral to ventromedial, so that distinct stimulation effects on interneurons and different types of principal neurons could be considered. Worth mentioning in this context is a tractographic study using local field potential recordings of dorsolateral and further ventral STN stimulation sites, which showed gradual connectivity changes in dorso-ventral direction, e.g., with respect to the additional motor cortex [29]. A recent diffusion tensor imaging tractography study underscored the involvement of white mater tracts connecting the stimulated STN tissue with the tegmentum, supplementary, premotor and primary motor cortex as well as the contralateral cerebellar hemisphere [99]. As is known from neurostimulation models in non-human animals, complex stimulation-related and neuroanatomical factors modulate the susceptibility of neighboring cell bodies and passing axons for co-stimulation [100]. Thus, given the small size of the STN and its close vicinity to various gray matter areas and white matter tracts [101], unintended co-stimulation particularly of the hypothalamus, substantia nigra, and the medial forebrain bundle is believed to causes cognitive and behavioral side effects [102, 103]. At the same time, more than 50% of VAT outside the STN was associated with good clinical outcomes, possibly by therapeutic co-stimulation of axonal tissue confining the STN dorsally, laterally and posteriorly [57]. Regarding language functions, a recent study in ten individual described a correlation between postoperatively improved semantic fluency and VAT outside the STN together with improved motor functions and VAT overlap with the motor STN [104]. Against this background, we would welcome an extension of the functional neuroanatomical basis of DBS effects on language by including tractography or functional connectivity measures in future studies.

Important to note in this context, although the associative STN is closely connected with cortical regions putatively supporting language production (i.e. monosynaptic connections with the prefrontal cortex [33, 34] and supplementary motor area [44]), the specific involvement of the STN in the language system is far from conclusive: whereas the above cortical areas [105–107] as well as thalamic and striatal areas [107, 108] have been proposed to interact with the "traditional" perisylvian language areas, functional network analyses provided no evidence of a corresponding engagement of the STN [107–109]. Other network studies, however, suggested an indirect involvement of the STN in language functions by means of word selection, motor planning, initiation, response inhibition, and action monitoring within the cortical-subcortical network [42, 110].

In the present study, faster language production was accompanied by a decrease exclusively in so-called non-linguistic pauses, i.e., hesitations, which typically interrupt the speech flow in PD patients [11]. Although this result did not reach the corrected level of significance, it points in the same direction as two previous studies, which reported an improved ratio of speech to non-speech pauses [55, 111] (but cf. [53]) and could thus indicate an amelioration of underlying word access and selection difficulties (cf. [112]). Importantly, the available data, although limited by the rather small sample size, seem to speak against a relevant effect of the articulation rate (at syllable level), which did not change significantly.

Our study did not indicate specific effects of STN DBS on the use of diverse lexemes or so-called open class words. Corresponding results could have been interpreted as an amelioration of reduced informative content described among PD patients [14, 18] and DBS-induced increases in noun and verb use have indeed been reported in an earlier study [56]. However, with regard to low effect sizes, the results should be interpreted with caution, as they may be due to the rather small sample size and do not necessarily indicate the absence of stimulation effects. Similarly, we found no significant DBS-related effects on sentence complexity, which can be reduced in PD patients [11] (but cf. [12]). However, as indicated by a very small effect size, also this result may be skewed by the small sample size and may not necessarily indicate the absence of stimulation effects.

Our findings regarding stylistic devices remained suggestive only: despite considerable effect sizes, ON-OFF differences did not reach the level of significance and high standard deviations limit the interpretability. However, there was an increase of so-called permutations including analogies, metaphors, ironies, and exaggerations, which require cognitive transfer between concrete expressions and their substitutes and could therefore relate to a DBS-induced thought flexibilization. Future studies investigating longer speech samples and a larger cohort may specifically tackle this question.

## Limitations of the study

The current study aimed to explore effects of therapeutic DBS on language parameters. It did not include experimental conditions, such as separated DBS testing per hemisphere or low-frequency stimulation, because we considered the testing to be an excessive strain for the participants and since we were interested in potential effects, as they occur in the clinical real-word scenario. Additionally, we focused on VAT overlap with STN subregions whose neuroanatomical segregation is known in detail, leaving aside complex effects due to potential co-stimulation of structures adjacent to the STN or through recruitment of the white matter tracts. Furthermore, by including a pre-operative condition, future studies could assess the possibility of active stimulation counteracting adverse effects of DBS implantation on language performance (cf. [113]). Also an inclusion of cognitive speed measures could be considered to verify the here proposed relation between cognitive and linguistic speed enhancements. Moreover, a possible bias towards participants who particularly profited from STN DBS in our cohort

cannot be excluded, given an earlier report on language benefits exclusively in patients with a left hemispheric disease dominance [56].

## Conclusion

In conclusion, therapeutic bilateral DBS of the STN accelerated clause and word production during natural language in PD patients. The magnitude of this increase was associated with the VAT within the associative STN, rostral to the dorsolateral motor section of the nucleus. The observed enhancements seem best compatible with the idea of STN DBS releasing executive procedures from excessive inhibition. Unexpectedly, this effect did not occur at the expense of language accuracy. Interestingly, corresponding accelerations appeared independent from prokinetic DBS effects, which correlated with a more prominent stimulation within the STN motor region. In sum, subtle variations in electrode localization may exert differential effects on motor and language functions, which warrant future efforts to expand on approaches for individualized and symptom-specific neuromodulation.

## Supporting information

**S1 Fig. Correlations between electric field overlap with associative vs. motor STN and language vs. motor changes.** Spearman correlations between the intersection of the electric field with each STN subregion (combined for both hemispheres) and changes in UPDRS, word and clause production rates, and error rate indicated A) a positive correlation between the improvement in the *UPDRS* and the electric field overlap with the motor STN, but not with the associative STN (top left), B) an association between the increase in the *word production rate* and the electric field overlap with the associative STN (on the level of trend), but not the motor STN (bottom left), and C) a positive correlation between the increase in the *clause production rate* and the electric field overlap with the associative STN, but not with the motor STN (bottom right). D) changes in error rates were neither significantly related to the electric field overlap with the associative, nor to the motor STN (top right).
(TIF)

**S1 Table. Electrode localization in the XYZ-space.** Overview of the localization of active electrodes in the xyz-space; due to missing post-operative MRI the data from participant no. 8 did not enter the localization analysis. Electrode localizations of all 26 active electrodes from the thirteen participants are depicted in Fig 1. Pt.: participant code.
(DOCX)

**S2 Table. Stylistic devices.** Overview of all semantic (A) and syntactic (B) stylistic devices analyzed, including the overall type and subcategories.
(DOCX)

**S3 Table. Language errors.** Distribution of language error rates (number of errors per total word count) across categories and participants.
(DOCX)

## Acknowledgments

We would like to thank all participants of this study.

## Author Contributions

**Conceptualization:** Felicitas Ehlen, Fabian Klostermann.

**Data curation:** Felicitas Ehlen.

**Formal analysis:** Felicitas Ehlen, Bassam Al-Fatly.

**Funding acquisition:** Fabian Klostermann.

**Investigation:** Felicitas Ehlen.

**Methodology:** Felicitas Ehlen, Fabian Klostermann.

**Project administration:** Fabian Klostermann.

**Resources:** Andrea A. Kühn, Fabian Klostermann.

**Supervision:** Andrea A. Kühn, Fabian Klostermann.

**Validation:** Andrea A. Kühn, Fabian Klostermann.

**Writing – original draft:** Felicitas Ehlen.

**Writing – review & editing:** Bassam Al-Fatly, Andrea A. Kühn, Fabian Klostermann.

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
