## [Decision Letter · Decision Letter 0]

14 Oct 2020

PONE-D-20-28761

Impact of deep brain stimulation of the subthalamic nucleus on natural language in patients with Parkinson’s disease

PLOS ONE

Dear Dr. Ehlen,

Thank you for submitting your manuscript to PLOS ONE. After careful consideration, we feel that it has merit but does not fully meet PLOS ONE’s publication criteria as it currently stands. Therefore, we invite you to submit a revised version of the manuscript that addresses the points raised during the review process.

We look forward to receiving your revised manuscript.

Kind regards,

Allan Siegel

Academic Editor

PLOS ONE

Journal Requirements:

Additional Editor Comments:

The present study was designed to determine whether electrical stimulation of the subthalamic nucleus (Sth) could reduce verbal errors that are associated with Parkinson’s disease (PD) in some patients. The study revealed two findings that are of interest and of potential significance. The first is that stimulation of the Sth could reduce verbal errors and the second suggested the possible presence of a functional localization within the Sth.

A number of concerns were raised by the reviewers and editor and they are briefly summarized here.

1. One concern is the relatively small sample size which weakens confidence in some of the conclusions reached by the authors (e.g., the discussion of negative results which could be softened somewhat).

2. The timing of the delivery of medication relative to when the tests were administered was not clear and should be indicated in the Methods Section as well as the rationale for selection of that time period.

3. The properties and characteristics of the electrodes need to be specified and the reasons why bipolar and monopolar electrodes were used. Each has its own characteristics regarding the extent to which current is spread at the sites of stimulation.

4. The frequency of stimulation was not specified. Apparently, this could be an important variable since it has been shown that changes in the frequencies applied can significantly alter the observed effects. The authors should also indicate why the frequency they selected was chosen,

5. Statements regarding the extent of current spread and regions activated by stimulation are likely to be speculative. Much of this work was conducted in animals a number of decades ago and provided a good amount of information detailing the spread of current with reference to the type of electrode utilized and amount of current applied. The authors should go back into the literature and see if they can apply such findings to the present study. I believe that those animal studies were far more precise than those recently obtained in human studies.

6. It is better to describe the stimulation applied in current values. (i.e., � Amp values) than in voltage.

7. The issue of functional localization within the Sth could be important and more discussion should be given to this section. There is evidence that there are at least two type of neurons located in dorsal and ventral aspects of the Sth that are immunoreactive for calretinin and parvalbumin (see Emmi, et al., Neuroanat., 22 April 2020). Such information could be helpful in expanding and clarifying this aspect of the Discussion. Regarding the authors’ attempts to draw conclusions about functional localization within the Sth, this structure is relatively small. So how can we eliminate the possibility that stimulation non-specifically activates widespread regions of Sth so as to cast doubt on a functional localization hypothesis? Further discussion of this matter would be helpful.

8. Some analysis or discussion of whether the error rates were cognitive or motor should be presented.

9. It is not clear how comparisons with what appears to be a control condition were made relative to the effects of stimulation. Were they taken before or after? Were the conditions randomized or systematically altered?

10. VTA is a common abbreviation for the ventral tegmental area. You might want to change this abbreviation to avoid any confusion.

11. I didn’t see any separate section showing figure captions. If this correct, then it should be added to the paper.

Other comments by both reviewers (not included above) are presumed to be included in this review and should be addressed as well.

Reviewers' comments:

Reviewer's Responses to Questions

**Comments to the Author**

1. Is the manuscript technically sound, and do the data support the conclusions?

Reviewer #1: Partly

Reviewer #2: Yes

2. Has the statistical analysis been performed appropriately and rigorously? 

Reviewer #1: No

Reviewer #2: Yes

3. Have the authors made all data underlying the findings in their manuscript fully available?

Reviewer #1: No

Reviewer #2: Yes

4. Is the manuscript presented in an intelligible fashion and written in standard English?

Reviewer #1: Yes

Reviewer #2: Yes

5. Review Comments to the Author

Reviewer #1: The authors present a nice, concise story of how STN DBS may affect language production in patients with Parkinson’s disease. They test patients with both stimulation on and off and see significant differences in word production, which correlates with VTA overlap of the associative region of the STN. Importantly, they also show that UPDRS change correlates better with VTA overlap of the motor STN, than associative, and that the reverse of these associations does not hold (no correlation with motor overlap and language and no correlation with associative overlap and UPDRS). My main concern is with the tone of the discussion on negative results. This is a very small sample of patients, so not finding an association is not proof that no association exists. Some sample language follows: “our results did not indicate a specific decrease in non-linguistic pauses and therefore rather argue for a general language acceleration.” “Further, the acceleration of language parameters was not explained by higher articulation rates, since these remained largely unchanged.” In both of these examples, there WERE differences, though they did not reach significance. That could either be because there is truly no association, or because the study is too small. So the authors need to soften their language—they shouldn’t make claims that are not statistically supported.

Some other issues follow:

p. 7: Did the authors control for timing of medication dosing between the two test periods? That is, if a patient was in an ON state following medication dosing while OFF DBS, vs. being in an OFF state while OFF DBS, that might change the results.

p.7: There is no electrode model #3378. Do the authors mean 3387?

p.8: The authors use a “heuristic” value (also known as “arbitrary” value) of 0.2 V/mm for defining their VTAs. Because their results depend so heavily on VTAs, they should do two things. One, provide references to support this choice. Two, perform a sensitivity analysis to see how much their results rely on this precise value, vs. being robust to an arbitrary choice like this.

P.13 and 14: A lot of these metrics are inherently correlated. For example, if you say more words per second, you will (all else being equal) say more clauses per second. So it’s not surprising that the two are correlated. What doesn’t make sense to me is how articulation (syllables per second) and pauses are NOT significantly different, while word production IS significantly different. If the patients are not saying more syllables or taking shorter pauses, how are they getting more words out?

I don’t see specifically how the data will be made available in the manuscript. This is a requirement for PLOS papers.

Reviewer #2: This is a thoughtful and careful analysis of speech symptoms of STN DBS. I found it to be well done and well-written with very helpful analyses of effect sizes and limitations. I have some minor questions:

1) One of the largest effects was the error rate in Table 2. Was this a cognitive or motor phenotype? Could this be influenced by cognitive control? Line 267 - the Bonferroni correct was applied, but I'm not sure if the Ho can be supported or rejected...

2) On line 235, authors refer to multiple comparisons correction, but don't say how. Later, the refer to Bonferroni, and it appears that 14 tests were included here. I can't track how this was done and why. Bonferroni can over-correct, so I wonder if FDR is a more helpful approach.

3) Would the frequency of DBS have mattered for speech? There is data that low-frequency DBS can affect cognitive control (PMID: 29203130 / PMID: 29190362) or even timing (PMID: 21931767) - discussing these papers may help the authors argument.

4) VTA may be confused with the ventral tegmental area. I wonder if different acronym would be clearer.

5) Line 154 - I think the Edlow's reference MRI was postmortem - but when I read this, I thought the patients in the present study had died. Clarify?

6) Did the authors study speech intelligibility?

7) Line 322 - I don't know what the 'sense of' means here. What is the evidence that STN DBS patients are disinhibited - and evidence in this study?

8) 331 - here proposed is awkward

9) Line 389 - I would remove the word fully

10) Is Figure 2B cut off? The position of the label made me unsure

11) How many patients trajectories are plotted in Fig 1A and locations in Fig 1B? How does this link to the table?

6. PLOS authors have the option to publish the peer review history of their article (what does this mean?). If published, this will include your full peer review and any attached files.

Reviewer #1: No

Reviewer #2: No

---

## [Author Response · Author response to Decision Letter 0]

28 Nov 2020

Dr. Felicitas Ehlen

Charité – Universitätsmedizin Berlin, Department of Neurology, Motor and Cognition Group 

Hindenburgdamm 30

12203 Berlin, Germany

Email: felicitas.ehlen@charite.de November 27, 2020

To

Professor Allan Siegel

Dear Professor Siegel,

Please find enclosed the revised version of our manuscript PONE-D-20-28761

„Impact of deep brain stimulation of the subthalamic nucleus on natural language in patients with Parkinson’s disease“ (marked-up and unmarked version) and the rebuttal letter, in which we address each issue in detail. We sincerely thank you and the reviewers for your thoughtful and constructive comments. We believe that the corresponding changes to the manuscript have led to significant improvements. Regarding compatibility aspects, we have created a detailed dataset, which is stored in the public repository "G-Node GIN" (https://gin.g-node.org/; URL: https://doi.gin.g-node.org/10.12751/g-node.mb83iq/; DOI: https://doi.org/10.12751/g-node.mb83iq). Also, we have used the digital diagnostic tool PACE to ensure that our images meet PLOS requirements, moved the captions for our supporting information to the end of the manuscript, and updated the in-text citations.

Thank you very much for reconsidering our manuscript. 

Sincerely, 

Felicitas Ehlen

Response to Reviewers

We would like to thank the editor and the reviewers for their thoughtful and constructive remarks. Due to conversion of the manuscript to PDF during the upload process, line numbers provided below may differ from those given in the PDF. 

Editor Comments:

1. One concern is the relatively small sample size which weakens confidence in some of the conclusions reached by the authors (e.g., the discussion of negative results which could be softened somewhat).

Reply: We address this issue critically in the revised version of the manuscript. The corresponding changes are given in detail in the reply to remark no. 1 by Reviewer #1.

2. The timing of the delivery of medication relative to when the tests were administered was not clear and should be indicated in the Methods Section as well as the rationale for selection of that time period.

Reply: We have added this important information to the Methods, Results, and Discussion. Please see our reply to remark no. 2 by Reviewer #1 for more details.

3. The properties and characteristics of the electrodes need to be specified and the reasons why bipolar and monopolar electrodes were used. Each has its own characteristics regarding the extent to which current is spread at the sites of stimulation.

Reply: All participants received therapeutic deep brain stimulation of the STN via implanted tetrapolar electrodes (Medtronics, Model 3387). In the revised version we have added further information on the electrode characteristics (i.e., cylindrical shape, contact height: 1.5 mm, diameter: 1.27 mm; lines 143-144). 

Stimulation parameters that can be adjusted individually to improve clinical symptoms are amplitude, impulse width, frequency, and polarity. The according parameters are provided in the manuscript as well as the electrode localization in the xyz-space. In the revised version of the manuscript we have added individual impedance values (in Ω) to Table 1 and we have included the information that stimulation was constant voltage-driven in all participants (line 107). These values represent standard values reported in the current literature on STN DBS (e.g., Paff et al., 2020). In general, the therapeutic stimulation parameters for STN DBS (derived largely by trial and error) are usually monopolar cathodic with stimulus amplitudes between 1 and 5 V, a pulse duration between 60 and 210 µs, and a stimulus frequency between 60 and 180 Hz (McIntyre et al., 2004; Koeglsperger et al., 2019).

In the current study the individual STN DBS settings had been optimized with respect to clinical parameters. We have added this information to the section “Participants” (lines 108-115).

The specifics regarding the extent of current spread resulting from monopolar vs. bipolar stimulation are considered by the modeling software model used in our study, which is now described in more detail in the Methods section (lines 168-184).

4. The frequency of stimulation was not specified. Apparently, this could be an important variable since it has been shown that changes in the frequencies applied can significantly alter the observed effects. The authors should also indicate why the frequency they selected was chosen.

Reply: All patients have been treated with high frequency stimulation. The individual frequency is provided in Table 1. Generally, therapeutic high frequency stimulation (i.e., between 60 and 180 Hz.; c.f. McIntyre et al., 2004; Koeglsperger et al., 2019) was applied, which is far above experimental low frequency stimulation (between 3 and 10 Hz; c.f. Wojtecki et al., 2006 & 2011; Kelley et al., 2018; Scangos et al. 2018). Regarding the procedure for selecting the individual frequency we would like to refer to the answer given to remark no. 3 above and we would like to refer to the answer given to remark no. 4 by Reviewer #2 below regarding the question of whether frequency of DBS may have mattered for speech.

5. Statements regarding the extent of current spread and regions activated by stimulation are likely to be speculative. Much of this work was conducted in animals a number of decades ago and provided a good amount of information detailing the spread of current with reference to the type of electrode utilized and amount of current applied. The authors should go back into the literature and see if they can apply such findings to the present study. I believe that those animal studies were far more precise than those recently obtained in human studies.

Reply: In the revised version of the manuscript we refer to a crucial review by J. B. Ranck (1975) on the heterogeneous effects electrical stimulation has on cell bodies and axons of the mammalian central nervous system (lines 449-451). Ranck not only reviewed relevant stimulation-related and neuroanatomical factors, but was also able to deduce an inverse relationship between the conduction velocity of an axon and the current required to stimulate it and suggested that “myelinated axons passing through an area and cell bodies with myelinated axons in an area are stimulated much more readily than any other elements”.

However, studies in nonhuman animal cannot provide information on current spread in humans since anatomy differs and electrode type and pulse configuration used in animal experiments differs from human DBS. Many groups have worked on VAT models to simulate the electrical field and tissue activated in patients. In vivo recordings of monkeys and rats treated with STN DBS showed both activating and inhibitory stimulation effects, varying in complex ways between local cell bodies and axons, afferent inputs, and fibers of passage (see McIntyre et al. 2004 for a review). For example, Hashimoto et al. (2003) achieved a voltage and frequency dependent motor improvement in two MPTP-lesioned rhesus monkeys treated with a scaled-down version of the DBS system used in the present study. Temel et al. (2005) investigated the effects of different stimulation currents on cognitive and motor functions in rats with bilateral STN DBS at varying amplitudes between 1, 3, 30 and 150 µA. The authors found amplitude-specific stimulation effects which were interpreted as an improvement of cognitive functions at 3 µA and of motor functions at 30 µA. They proposed that STN DBS affects cognitive and motor performance differently as a function of stimulation current and suggested “that each loop has unique electrical properties which can be modulated independently”. However, no recordings were made from the cerebral structures involved, so the proposed effects of current spread remain speculative. Furthermore, as stated by Spieles-Engemann (2010), rat models are somewhat limited due to the comparatively small size of the rat STN relative to the DBS electrode, so that the VAT is expected to exceed that in humans. 

Although these and similar results from animal experiments are essential for the understanding of STN DBS, we did not find any literature specifically addressing current spread in the motor and associative parts of the STN (or adjacent areas) in non-human animals. However, based on recent studies in humans with STN DBS, we now discuss the question of current propagation in more detail (lines 451-459).

With regard to the algorithms currently used to estimate current spread associated with therapeutic STN DBS in humans, it seems important to note that advanced models have been developed over the last decade to calculate the volume of activated tissue (VAT). The VAT is conceptualized as the field activated by the DBS due to the additional stimulation of myelinated axons. Butson et al. (2006) first presented a model to predict the VAT by coupling finite element models of the electrodes and surrounding medium to cable models of myelinated axons, followed by a model system for identifying the intersection between the VAT and the surrounding structures, which was obtained from a combination of an MRI-based brain model, a finite element model of the electrode and its electric field, and a VAT prediction derived from the response of myelinated axons to the applied electric field as a function of stimulation parameters (Butson et al., 2007). Maks et al. (2009) integrated neuroimaging, neurophysiological, and stimulation data to determine the expected VAT in patients treated with STN DBS in relation to therapeutic stimulation parameters (i.e. contact, impedance, amplitude, frequency, pulse width). Under these assumptions, their model indicated an activation of approximately 70 mm3 of axonal tissue dorsally, laterally and posterior to the center of the STN.

These and related approaches are considered to deliver relatively accurate predictions of the VAT (Chaturvedi et al., 2013). However, limitations include, e.g., an overestimation of its size (Butson et al., 2006). Further refinements to calculate the VAT came from artificial neural networks aiming to improve the predictions in the case of bipolar or multipolar stimulation (Chaturvedi et al., 2013).

The software used in the current study (Lead-DBS Version 2; Horn et at. 2019) is an established option for modeling the VAT in patients treated with DBS. To estimate the VAT, the software designers used existing models (first proposed by Kuncel et al., 2008; Mädler et al. 20012; Dembek et al. 2017; Horn et al., 20017) as well as tractography algorithms to transition from local to global modulation volume. On this basis, the program calculates the assumed VAT based on parameters related to the location of the active electrode, the lead and the adjacent gray and white matter (on an atlas basis) as well as clinical stimulation settings. Under these premises we believe that our approach—despite its limitations—is sufficiently accurate and represents the current state of the art. We have extended our description of this method in the revised version of the manuscript (lines 168-184).

6. It is better to describe the stimulation applied in current values. (i.e., � Amp values) than in voltage.

Reply: All participants were treated with constant voltage-driven stimulation. In this case current values are not provided by the DBS recording system. Since voltage is held constant, the current flow is determined by the impedance of the DBS electrode at the contact-tissue interface so that clinicians and researchers cannot directly control or measure the current flow (Butson et al. 2008; Paff et al. 2020). In the revised version of the manuscript we therefore provide individual impedance values in Table 1. 

7. The issue of functional localization within the Sth could be important and more discussion should be given to this section. There is evidence that there are at least two type of neurons located in dorsal and ventral aspects of the Sth that are immunoreactive for calretinin and parvalbumin (see Emmi, et al., Neuroanat., 22 April 2020). Such information could be helpful in expanding and clarifying this aspect of the Discussion. 

Regarding the authors’ attempts to draw conclusions about functional localization within the Sth, this structure is relatively small. So how can we eliminate the possibility that stimulation non-specifically activates widespread regions of Sth so as to cast doubt on a functional localization hypothesis? Further discussion of this matter would be helpful.

Reply: Thank you for this important advice. We have extended our Discussion and now refer to the results of Levesque and Parent (2005) regarding the distribution of different cell populations within the STN, which was embraced by Emmi et al. (2020). We recently showed a frequency-specific distribution of predominant synchronized neuronal activity within the STN that may underline a functional subdivision of the nucleus (Accolla et al., 2016; Horn et al., 2017). Accordingly, these areas are predominantly involved in different but overlapping cortico-subcortical functional networks (lines 443-446). Against this background, we would welcome an extension of the functional neuroanatomical basis of DBS effects on language by including tractography or functional connectivity measures in future studies.

8. Some analysis or discussion of whether the error rates were cognitive or motor should be presented.

Reply: The term "error" was used to refer to language errors. To make this clear, we have changed the term "error" to " language error" throughout the manuscript. For a more detailed answer, we would like to refer to remark no. 1 by Reviewer #2.

9. It is not clear how comparisons with what appears to be a control condition were made relative to the effects of stimulation. Were they taken before or after? Were the conditions randomized or systematically altered?

Reply: In the present study, we evaluated the natural language performance of 14 consecutive participants with Parkinson's disease chronically treated with STN DBS and compared their respective performance in the ON vs. OFF DBS condition. Therefore, each participant was examined twice, once in the STN DBS ON and once in the STN DBS OFF condition. The two sessions were held at two-month intervals, systematically alternating the order of ON and OFF examinations across the participants. This is now expressed more clearly in the Methods section (lines 128-129).

10. VTA is a common abbreviation for the ventral tegmental area. You might want to change this abbreviation to avoid any confusion.

Reply: We have changed the term VTA to VAT (as used, e.g., by Mosley et al, 2020) throughout the manuscript.

11. I didn’t see any separate section showing figure captions. If this correct, then it should be added to the paper.

Reply: Following the PLOS ONE style templates, the figure captions are provided in the text body immediately after the labels “Fig 1.” and “Fig 2.”.

Reviewer #1: 

1. My main concern is with the tone of the discussion on negative results. This is a very small sample of patients, so not finding an association is not proof that no association exists. Some sample language follows: “our results did not indicate a specific decrease in non-linguistic pauses and therefore rather argue for a general language acceleration.” “Further, the acceleration of language parameters was not explained by higher articulation rates, since these remained largely unchanged.” In both of these examples, there WERE differences, though they did not reach significance. That could either be because there is truly no association, or because the study is too small. So the authors need to soften their language—they shouldn’t make claims that are not statistically supported.

Reply: In accordance with your advice, we now present non-significant findings more cautiously and mention small effect sizes as an indicator of a possibly insufficient sample size. The following changes have been made:

• “Further semantic and syntactic language features remained vastly unchanged.” was changed to: “No significant changes in further semantic or syntactic language features were detected in the current study”. (lines 51-52)

• “No relevant ON-OFF differences were indicated for speech pauses, sentence complexity, the use of open class words or diverse lexemes.” was changed to: “No significant ON-OFF differences were detectable for sentence complexity, the use of open class words or diverse lexemes, or the total duration of linguistic pauses, each showing rather low effect sizes”. (lines 372-375)

• “…our results did not indicate a specific decrease in non-linguistic pauses and therefore rather argue for a general language acceleration.” Has been changed completely due to a change of the parameter (please see remark no. 5 below). The section now reads: “In the present study, faster language production was accompanied by a decrease exclusively in so-called non-linguistic pauses, i.e., hesitations, which typically interrupt the speech flow in PD patients [11]. Although this result did not reach the corrected level of significance, it points in the same direction as two previous studies, which reported an improved ratio of speech to non-speech pauses [55,111] (but cf. [53]) and could thus indicate an amelioration of underlying word access and selection difficulties (cf. [112])”.

• “Further, the acceleration of language parameters was not explained by higher articulation rates, since these remained largely unchanged” was changed to: “Importantly, the available data, although limited by the rather small sample size, seem to speak against a relevant effect of the articulation rate (at syllable level), which did not change significantly”. (lines 478-480)

• We have added “However, with regard to low effect sizes, the results should be interpreted with caution, as they may be due to the rather small sample size and do not necessarily indicate the absence of stimulation effects” to the sentence “Our study did not indicate specific effects of STN DBS on the use of diverse lexemes or so-called open class words”. (lines 484-487)

• We have changed “Similarly, we found no DBS-related effects on sentence complexity, which can be reduced in PD patients [ref.]” to “Similarly, we found no significant DBS-related effects on sentence complexity, which can be reduced in PD patients [ref.]. However, as indicated by a very small effect size, also this result may be skewed by the small sample size and may not necessarily indicate the absence of stimulation effects”. (lines 487-490)

2. p. 7: Did the authors control for timing of medication dosing between the two test periods? That is, if a patient was in an ON state following medication dosing while OFF DBS, vs. being in an OFF state while OFF DBS, that might change the results.

Reply: The timing of the tests had been planned individually to ensure the best clinical “medication-ON” state. This resulted in a testing period between 11 a.m. and about 1 p.m. for most participants. The movement state was checked clinically prior to switching of off the stimulation by asking the participant about their present subjective state and observing their movements (UPDRS was, unfortunately not assessed before switching off). The last intake of their medication (and dosage) was protocolled. There was no significant difference in timing between the last intake of their medication and the beginning of the language assessment (ON: 1.64 ± 1.434 hours; OFF: 2.286 ± 1.369 hours; p-value: 0.089; Cohen’s d: 0.459). We have added this information to the Methods (lines 134-136) and the Results (lines 303-306).

3. p.7: There is no electrode model #3378. Do the authors mean 3387?

Reply: Thank you. This error has been corrected (line 143).

4. p.8: The authors use a “heuristic” value (also known as “arbitrary” value) of 0.2 V/mm for defining their VTAs. Because their results depend so heavily on VTAs, they should do two things. One, provide references to support this choice. Two, perform a sensitivity analysis to see how much their results rely on this precise value, vs. being robust to an arbitrary choice like this.

Reply: 

1. The value of 0.2 V/mm was originally used by Hemm at al. (2005), validated by Åström et al. (2015), and used in similar study designs (e.g., Horn et al., 2017 and Irmen et al., 2019). We now provide these references in the Methods (line 182).

2. To assess whether the results are dependent on the 0.2 V/mm threshold, we performed a control analysis that did not include a heuristic/arbitrary threshold, but instead used the full electric field (the intersection of the electric field with each STN subregion was calculated bilaterally and the electric field strengths of the intersections of both hemispheres were combined, see S1 Fig.). This approach largely verified our results by showing: 

a. improvement in the UPDRS was positively correlated with the electric field overlap with the motor STN (r = 0.57, p = 0.022), but not with the associative STN (r = 0.36, p = 0.10),

b. increase in the word production rate was associated with the electric field overlap with the associative STN (r = 0.41, p = 0.07; thus not statistically significant, but on the level of trend), but not with the motor STN (r = 0.10, p = 0.34),

c. increase in the clause production rate was correlated with electric field overlap with the associative STN (r = 0.71, p = 0.001), but not with the motor STN (r = 0.14, p = 0.32),

d. changes in error rates were neither significantly related to the electric field overlap with the associative (r = 0.13, p = 0.35), nor the motor STN (r = 0.35, p = 0.11).

This information is now provided in the Methods (lines 185-188) and the supporting information (lines 859-869).

5. P.13 and 14: A lot of these metrics are inherently correlated. For example, if you say more words per second, you will (all else being equal) say more clauses per second. So it’s not surprising that the two are correlated. What doesn’t make sense to me is how articulation (syllables per second) and pauses are NOT significantly different, while word production IS significantly different. If the patients are not saying more syllables or taking shorter pauses, how are they getting more words out?

Reply: Thank you for pointing this out. Instead of the former use of the ratio of linguistic to non-linguistic pauses, we have now included the primary values “total duration of linguistic pauses” and “total duration of non-linguistic pauses”, which allow for a better interpretation of the otherwise inconclusive value. According changes have been made in lines 234-235, 271-272, 315, 372, 374, 473-474, and in Table 2. Further pause-related parameters can also be found in minimal data set (i.e., mean and total pause duration, the number and duration of linguistic and non-linguistic pauses; columns AU-BA). The data show a shorter duration of non-linguistic pauses (ON: 6.518 ± 3.149 s; OFF: 10.020 ± 6.221 s; p = 0.049; Cohen's d: 0.710) in the ON vs. OFF condition, while the duration of linguistic pauses was vastly unchanged (ON: 16.076 ± 7.942 s; OFF: 16.273 ± 9.330 s; p = 0.946; Cohen's d: 0.023). Despite the statical non-significance after controlling for the false discovery rate, this finding points towards an acceleration of language production due to shorter non-linguistic pauses while linguistic pauses and the articulation rate of each single word did not seem to have relevantly affected the result. 

6. I don’t see specifically how the data will be made available in the manuscript. This is a requirement for PLOS papers.

Reply: We have created a minimal data set which is stored publicly at G-Node GIN (https://gin.g-node.org/; URL: https://doi.gin.g-node.org/10.12751/g-node.mb83iq/; DOI: https://doi.org/10.12751/g-node.mb83iq). This is now mentioned in the Methods (line 252). In addition to the tables presented in the manuscript and the supplement, the minimal data set includes detailed individual (de-identified) data belonging to the following categories, which have been analyzed in the current study: 

• PANDA Score, UPDRS and Hohn&Yahr Stage 

• Medication

• Basic Data of the Monologs

• Pauses

• Word Production Rate

• Language Errors

• Word Classes

• Constituents

• Type Token Ratio

• Morphosyntactic Categorizations

• Sentences and Clauses

• Sentence Complexity

• Syntactic and Semantic Stylistic Devices 

Reviewer #2: 

1. One of the largest effects was the error rate in Table 2. Was this a cognitive or motor phenotype? Could this be influenced by cognitive control? 

Reply: Thank you for addressing this question. Our error rate refers to language errors (we have now changes the term „errors“ to „language errors“ throughout the manuscript). When rating language errors we referred to the linguistic categories, i.e., 1. grammatical, 2. lexical, 3. phonetic, 4. contextual, 5. stylistic, 6. idiomatic, 7. pragmatic, 8. logic (lines 257-258). Since the errors were distributed rather heterogeneously across categories and participants (see table below, which has also been added to the supplement as S3 Table), we decided to assess the combined error rate rather than the subcategories. On a descriptive level, however, it can be said that most errors belonged to the categories grammatical, stylistic, and pragmatic. 

 DBS ON DBS OFF 

Errors per total word count Mean SD Mean SD

Grammatical 0.050 0.034 0.063 0.034

Lexical 0.016 0.012 0.016 0.015

phonetic 0.005 0.012 0.004 0.008

contextual 0.000 0.000 0.000 0.000

Stylistic 0.020 0.023 0.050 0.043

idiomatic 0.002 0.003 0.001 0.003

Pragmatic 0.025 0.020 0.020 0.019

Logic 0.004 0.005 0.006 0.012

With respect to the discrepancy between a typically described increase in commission errors and improved error rates in the present study, our findings point towards a relevant difference between the natural language task and explicit cognitive tasks. The latter impose a significant cognitive load on the participants—particularly by demanding speed increases, decision making or no-go strategies—which are assumed to be sensitive to STN DBS-related disturbance of inhibitory control (Hershey et al., 2004; Frank et al., 2007; Houvenaghel et al., 2016; Neumann et al., 2018). We therefore suggest that STN DBS had beneficial effects on language accuracy, which, in the context of the literature on cognitive errors, appear rather independent from DBS-related inference with inhibitory control. Future studies could profit from including interventions to evaluate the effects of increased cognitive demands on the different types of language errors. This is now expressed more clearly in the Discussion, where we have also integrated the results of recent studies investigating differential effects of STN DBS on cognitive control (lines 412-429). 

2. Line 267 - the Bonferroni correct was applied, but I'm not sure if the Ho can be supported or rejected...

Reply: The entire section on language error rates has been changed after application of the FDR as suggested in the following remark. The sentence in question has therefore been deleted. 

3. On line 235, authors refer to multiple comparisons correction, but don't say how. Later, the refer to Bonferroni, and it appears that 14 tests were included here. I can't track how this was done and why. Bonferroni can over-correct, so I wonder if FDR is a more helpful approach.

Reply: Thank you for this advice. In the revised version of the manuscript we have determined the false discovery rate using the Benjamini–Hochberg procedure. This has been altered in the Methods (lines 275-278) and and in Table 2 of the Results section.

In the original version of the manuscript we had used the Bonferroni procedure in that we had adjusted the p-value to 0.007 (i.e., 0.05 divided by 7; due to seven ON-OFF comparisons: 1 = word production rate, 2 = types of speech pauses, 3 = error rate, 4 = clause production rate, 5 = sentence complexity, 6 = rate of open class words, 7 = type-token ratio). Please note that the revised version of the manuscript contains eight instead of seven parameters of interest (this is because we now assessed the total duration of linguistic pauses and of non-linguistic pauses instead of the ratio of linguistic to non-linguistic pauses; see reply given to remark no. 5 by Reviewer #1). 

Whereas in our original version the ON-OFF difference of the language error rate did not reach the corrected p-value, it can be considered significant according to the Benjamini–Hochberg procedure. In line with our evaluations of the other significant ON-OFF differences, we have correlated the language error reduction with the VTA overlap with the motor and associative area of the STN and with ON-OFF differences of the UPDRS and PANDA scores. We found no significant correlations. Corresponding changes to the manuscript have been made in the Abstract (lines 50-52), Results section ( lines 313, 345-347), and the Discussion (lines 364-365, 371-372, 430-433).

4. Would the frequency of DBS have mattered for speech? There is data that low-frequency DBS can affect cognitive control (PMID: 29203130 / PMID: 29190362) or even timing (PMID: 21931767) - discussing these papers may help the authors argument

Reply: We have now included a corresponding section in the Discussion, which reads: “In this framework it is of interest that experimental low frequency stimulation of the ventral STN at 5 Hz improved response control in a Stroop task [92] and dorsolateral stimulation at 4 Hz appeared to normalize the response latency in an interval timing task [33], which deteriorated at 10 Hz stimulation [93]. The authors therefore proposed a functional coupling between the prefrontal cortex and the STN in the delta/theta frequency range, which could be involved in cognitive control processes, including cue processing, response inhibition, working memory recruitment, and attention [33,92]. Having said that, improved cognitive parameters also indicated a possible activation of fronto-subthalamic connections when stimulated at 10 Hz. [94]. Overall, these results are compatible with the idea of a relevant role of the prefrontal subthalamic signaling pathway, which can be disturbed or compensated by co-stimulation of the associative part of the STN or its connective fibers, depending on stimulation parameters, localization and the cognitive requirements of the task type” (lines 399-411).

Given the therapeutic STN DBS frequency range between 60 and 180 Hz (both reported in the literature, e.g., McIntyre et al., 2004; Koeglsperger et al., 2019) and applied in the current study), frequencies of 10 or 4 Hz are considered very low and have been shown to worsen motor performance (Timmermann et al. 2004; Wojtecki et al. 2006). In our study we did not investigate experimental stimulation frequencies (this is now mentioned as a limitation, lines 501-502) and the four different therapeutic stimulation frequencies were distributed very heterogeneously among the participants (see table below). We have therefore not included the following analysis in the study but would like to present the results here in response to your question.: 

In order to estimate effects of stimulation frequencies, we used a linear mixed model with the fixed factor “stimulation frequency” and the dependent parameters “ON-OFF difference in error rates”, “ON-OFF difference in word production rates”, and “ON-OFF difference in clause production rates”. The model indicated no significant effects of the stimulation frequency on the three dependent variables (see table below). Due to the heterogeneous distribution of stimulation frequencies and very large standard deviations we believe the results to be of limited explanatory value. Although there seemed to be a positive association between the stimulation frequency and the increase in word and sentence production rates, Spearman-Rho tests indicated no statically significant correlation between frequencies and the ON-OFF difference in error rates (r = 0.27; p = 0.34), word production rates (r = 0.38; p = 0.18), or clause production rates (r = 0.04; p = 0.89).

 ON-OFF difference

 error rates word production rates clause production rates

 F3,10 = 1.45; p = 0.29 F3,10 = 0.61; p = 0.62 F3,10 = 0.03; p = 0.99

Stimulation Frequency cases (abs. and %) Mean SD Mean SD mean SD

80 1 (7.14%) 2.71 -3.27 36.66 

90 4 (28.57%) -55.89 19.68 16.63 19.72 45.32 54.88

130 8 (57.14%) -18.73 40.8 28.24 26.32 55.34 87.04

160 1 (7.14%) 1.15 29.09 55.45 

5. VTA may be confused with the ventral tegmental area. I wonder if different acronym would be clearer.

Reply: We have changed the term VTA to VAT (as used, e.g., by Mosley et al, 2020) throughout the manuscript.

6. Line 154 - I think the Edlow's reference MRI was postmortem - but when I read this, I thought the patients in the present study had died. Clarify?

Reply: We have corrected and split this ambiguous sentence into: „Next, each participant’s bilateral VATs were overlapped with the motor and associative area of the STN as provided by the DISTAL atlas” (lines 182-184) and “Backdrop is from the high resolution 100 micron postmortem MRI in Edlow et al. [72]. Please note that due to missing post-operative MRI the data from participant no. 8 did not enter the localization analysis.” (lines 196-198, i.e., caption of Figure 1).

7. Did the authors study speech intelligibility?

Reply: The recorded data of all participants included in the present study was generally well intelligible (recorded data from one person had to be excluded prior to any further assessment due to unintelligible speech in both stimulation conditions). Intelligibility was scored according to the UPDRS speech item. In so doing, the examiner evaluated volume, modulation, and clarity on a given scale (i.e., 0: normal: no speech problems. 1: slight: loss of modulation, diction, or volume, but still all words easy to understand. 2: mild: loss of modulation, diction, or volume, with a few words unclear, but the overall sentences easy to follow. 3: moderate: speech is difficult to understand to the point that some, but not most, sentences are poorly understood. 4: severe: most speech is difficult to understand or unintelligible; Goetz et al., 2008).

Wilcoxon signed-rank test indicated speech intelligibility to be vastly similar in both stimulation conditions (ON median: 1 point; OFF median: 1 point; p = 0.656; z < 0.001). This information has been added to the Methods (lines 126-127, 206-209) and Results (lines 302-303) sections. 

Speech intelligibility was, however not focused on in more detail in the present study (but in an earlier study of our research group; Klostermann et al., 2008) 

8. Line 322 - I don't know what the 'sense of' means here. What is the evidence that STN DBS patients are disinhibited - and evidence in this study?

Reply: The term "disinhibitory effects" was wrongly used in our manuscript in this context. We have corrected this mistake. The sentence now reads: “The main finding of increased clause and word production rates seems compatible with the idea that the STN accelerates cognitive speed...” (lines 376-377). We have furthermore added findings from other studies regarding ‘disinhibition’ to the discussion on error rates (please see also the reply given to remark no. 1). 

9. 331 - here proposed is awkward

Reply: We have changed the expression to: “Importantly, this enhancement seemed vastly unrelated to the DBS-induced motor improvement,...” 

10. Line 389 - I would remove the word fully

Reply: Thank you. This has been corrected.

11. Is Figure 2B cut off? The position of the label made me unsure

Reply: The original Figure 2 was not cut off. In the revised version we have moved the position of the labels „A“ and „B“ to the top, so the figure will not appear as if it had been cut off. 

12. How many patients trajectories are plotted in Fig 1A and locations in Fig 1B? How does this link to the table?

Reply: Trajectories and active electrodes are plotted from thirteen participants (due to missing post-operative MRI the data from participant no. 8 did not enter the localization analysis). In figure 1B right hemispheric electrodes have been projected onto the left STN for visualizing purposes (leading to a total of 26 active contacts). We have added this information to the caption of figure 1, which now reads: “A: DBS electrodes localizations from thirteen participants in relation to the STN. Electrodes have been reconstructed in MNI space. Red contacts represent clinically active contacts. B: Spatial distribution of active contacts in relation to different STN segments (from DISTAL atlas). All 26 active contacts (red spheres) are projected cumulatively onto the left hemisphere (right contacts have been flipped)...” (lines 190-195).

The s1-table provides the corresponding data on the localization of the active electrodes in the xyz-space of the thirteen participants in the left and right STN, which are depicted in Figure 1. We have added the sentence “Electrode localizations of all 26 active electrodes from the thirteen participants are depicted in figure 1” to the s1-table caption (lines 850-851). 

References used in the Response to the Reviewers:

Accolla EA, Herrojo Ruiz M, Horn A, Schneider GH, Schmitz-Hübsch T, Draganski B, et al. Brain networks modulated by subthalamic nucleus deep brain stimulation. Brain. 2016;139(Pt 9):2503–15. 

Butson CR, McIntyre CC. Role of electrode design on the volume of tissue activated during deep brain stimulation. J Neural Eng. 2006;3(1):1–8. 

Butson CR, Cooper SE, Henderson JM, McIntyre CC. Patient-specific analysis of the volume of tissue activated during deep brain stimulation. Neuroimage. 2007;34(2):661–70. 

Butson CR, McIntyre CC. Current steering to control the volume of tissue activated during deep brain stimulation. Brain Stimul. 2008;1:7–15. 

Chaturvedi A, Luján JL, McIntyre CC. Artificial neural network based characterization of the volume of tissue activated during deep brain stimulation. J Neural Eng. 2013;10(5):1–17.

Dembek TA, Barbe MT, Åström M, Hoevels M, Visser-Vandewalle V, Fink GR, et al. Probabilistic mapping of deep brain stimulation effects in essential tremor. NeuroImage Clin. 2017;13:164–173.

Frank MJ, Samanta J, Moustafa AA, Sherman SJ. Hold your horses: Impulsivity, deep brain stimulation, and medication in Parkinsonism. Science (80- ). 2007;318(5854):1309–12.

Goetz CG, Tilley BC, Shaftman SR, Stebbins GT, Fahn S, Martinez-Martin P, et al. Movement Disorder Society-Sponsored Revision of the Unified Parkinson’s Disease Rating Scale (MDS-UPDRS): Scale presentation and clinimetric testing results. Mov Disord. 2008.

Hashimoto T, Elder CM, Okun MS, Patrick SK, Vitek JL. Stimulation of the subthalamic nucleus changes the firing pattern of pallidal neurons. J Neurosci. 2003;23(5):1916 –1923.

Hershey T, Revilla FJ, Wernle A, Gibson PS, Dowling JL, Perlmutter JS. Stimulation of STN impairs aspects of cognitive control in PD. Neurology. 2004;62:1110–1114. 

Horn A, Neumann WJ, Degen K, Schneider GH, Kühn AA. Toward an electrophysiological “Sweet spot” for deep brain stimulation in the subthalamic nucleus. Hum Brain Mapp. 2017;38(7):3377–3390.

Horn A, Reich M, Vorwerk J, Li N, Wenzel G, Fang Q, et al. Connectivity Predicts deep brain stimulation outcome in Parkinson disease. Ann Neurol. 2017;82(1):67–78. 

Horn A, Li N, Dembek TA, Kappel A, Boulay C, Ewert S, et al. Lead-DBS v2: Towards a comprehensive pipeline for deep brain stimulation imaging. Neuroimage. 2019;184:293–316.

Houvenaghel JF, Duprez J, Argaud S, Naudet F, Dondaine T, Robert GH, et al. Influence of subthalamic deep-brain stimulation on cognitive action control in incentive context. Neuropsychologia. 2016;91:519–530.

Kelley R, Flouty O, Emmons EB, Kim Y, Kingyon J, Wessel JR, et al. A human prefrontal-subthalamic circuit for cognitive control. Brain. 2018;141(1):205–16. 

Klostermann F, Ehlen F, Vesper J, Nubel K, Gross M, Marzinzik F, et al. Effects of subthalamic deep brain stimulation on dysarthrophonia in Parkinson’s disease. J Neurol Neurosurg Psychiatry. 2008;79(5). 

Koeglsperger T, Palleis C, Hell F, Mehrkens JH, Bötzel K. Deep brain stimulation programming for movement disorders: Current concepts and evidence-based strategies. Front Neurol. 2019;10(Article 410):1–20. 

Kuncel AM, Cooper SE, Grill WM. A method to estimate the spatial extent of activation in thalamic deep brain stimulation. Clin Neurophysiol. 2008;119:2148–2158. 

Mädler B, Coenen VA. Explaining clinical effects of deep brain stimulation through simplified target-specific modeling of the volume of activated tissue. Am J Neuroradiol. 2012;33(6):1072–80.

Maks CB, Butson CR, Walter BL, Vitek JL, McIntyre CC. Deep brain stimulation activation volumes and their association with neurophysiological mapping and therapeutic outcomes. J Neurol Neurosurg Psychiatry. 2009;80(6):659–66.

McIntyre CC, Savasta M, Walter BL, Vitek JL. How Does Deep Brain Stimulation Work? Present Understanding and Future Questions. J Clin Neurophysiol. 2004;21:40–50.

Neumann WJ, Schroll H, De Almeida Marcelino AL, Horn A, Ewert S, Irmen F, et al. Functional segregation of basal ganglia pathways in Parkinson’s disease. Brain. 2018;141(9):2655–69.

Paff M, Loh A, Sarica C, Lozano AM, Fasano A. Update on current technologies for deep brain stimulation in parkinson’s disease. J Mov Disord. 2020;13(3):185–98.

Ranck JB. Which elements are excited in electrical stimulation of mammalian central nervous system: A review. Brain Res. 1975;98:417–40. 

Scangos KW, Carter CS, Gurkoff G, Zhang L, Shahlaie K. A pilot study of subthalamic theta frequency deep brain stimulation for cognitive dysfunction in Parkinson’s disease. Brain Stimul. 2018;11:456–8.

Spieles-Engemann AL, Collier TJ, Sortwell CE. A functionally relevant and long-term model of deep brain stimulation of the rat subthalamic nucleus: Advantages and considerations. Eur J Neurosci. 2010;32:1092–1099.

Temel Y, Visser-Vandewalle V, Aendekerk B, Rutten B, Tan S, Scholtissen B, et al. Acute and separate modulation of motor and cognitive performance in parkinsonian rats by bilateral stimulation of the subthalamic nucleus. Exp Neurol. 2005;193(1):43–52.

Timmermann L, Wojtecki L, Gross J, Lehrke R, Voges J, Maarouf M, et al. Ten-hertz stimulation of subthalamic nucleus deteriorates motor symptoms in Parkinson’s disease. Mov Disord. 2004;19(11):1328–33.

Wojtecki L, Timmermann L, Jörgens S, Südmeyer M, Maarouf M, Treuer H, et al. Frequency-dependent reciprocal modulation of verbal fluency and motor functions in subthalamic deep brain stimulation. Arch Neurol. 2006;63(9):1273–6.

Wojtecki L, Elben S, Timmermann L, Reck C, Maarouf M, Jörgens S, et al. Modulation of human time processing by subthalamic deep brain stimulation. PLoS One. 2011;6(9):1–10

---

## [Decision Letter · Decision Letter 1]

4 Dec 2020

Impact of deep brain stimulation of the subthalamic nucleus on natural language in patients with Parkinson’s disease

PONE-D-20-28761R1

Dear Dr. Ehlen,

Both reviewers and myself agree that you and your team have addressed the questions raised and made all the appropriate changes in the manuscript. Accordingly, we are pleased to inform you that your manuscript has been judged scientifically suitable for publication and will be formally accepted for publication once it meets all outstanding technical requirements.

Kind regards,

Allan Siegel

Academic Editor

PLOS ONE

Additional Editor Comments (optional):

Reviewers' comments:

Reviewer's Responses to Questions

**Comments to the Author**

1. If the authors have adequately addressed your comments raised in a previous round of review and you feel that this manuscript is now acceptable for publication, you may indicate that here to bypass the “Comments to the Author” section, enter your conflict of interest statement in the “Confidential to Editor” section, and submit your "Accept" recommendation.

Reviewer #1: All comments have been addressed

Reviewer #2: All comments have been addressed

2. Is the manuscript technically sound, and do the data support the conclusions?

Reviewer #1: Yes

Reviewer #2: Yes

3. Has the statistical analysis been performed appropriately and rigorously? 

Reviewer #1: Yes

Reviewer #2: Yes

4. Have the authors made all data underlying the findings in their manuscript fully available?

Reviewer #1: Yes

Reviewer #2: Yes

5. Is the manuscript presented in an intelligible fashion and written in standard English?

Reviewer #1: Yes

Reviewer #2: Yes

6. Review Comments to the Author

Reviewer #1: The authors have addressed my concerns. No further issues have been identified. Everything looks ok for publication.

Reviewer #2: I found the manuscript improved. I have no further comments. This will be an advance on our understanding of DBS.

7. PLOS authors have the option to publish the peer review history of their article (what does this mean?). If published, this will include your full peer review and any attached files.

Reviewer #1: **Yes: **John Rolston

Reviewer #2: No

---

## [Editor Report · Acceptance letter]

17 Dec 2020

PONE-D-20-28761R1 

Impact of deep brain stimulation of the subthalamic nucleus on natural language in patients with Parkinson’s disease 

Dear Dr. Ehlen:

I'm pleased to inform you that your manuscript has been deemed suitable for publication in PLOS ONE. Congratulations! Your manuscript is now with our production department. 

Kind regards, 

on behalf of

Dr Allan Siegel 

Academic Editor

PLOS ONE